# Influence of Dietary Polar Lipid Supplementation on Memory and Longitudinal Brain Development

**DOI:** 10.3390/nu13082486

**Published:** 2021-07-21

**Authors:** Joanne E. Fil, Sangyun Joung, Jonas Hauser, Andreas Rytz, Courtney A. Hayes, Ryan N. Dilger

**Affiliations:** 1Neuroscience Program, University of Illinois, Urbana, IL 61801, USA; jfil2@illinois.edu (J.E.F.); sjoung2@illinois.edu (S.J.); 2Société des Produits Nestlé, 1000 Lausanne, Switzerland; jonas.hauser@rdls.nestle.com (J.H.); andreas.rytz@rdls.nestle.com (A.R.); 3College of Veterinary Medicine, University of Illinois, Urbana, IL 61801, USA; ahern@illinois.edu; 4Department of Animal Sciences, University of Illinois, Urbana, IL 61801, USA; 5Division of Nutritional Sciences, University of Illinois, Urbana, IL 61801, USA

**Keywords:** brain, neurodevelopment, magnetic resonance imaging, behavior, Gompertz, longitudinal, polar lipids

## Abstract

Polar lipids, which are found in human milk, serve essential functions within biological membranes, hence their importance in brain development and cognition. Therefore, we aimed to evaluate the longitudinal effects on brain macrostructural and microstructural development and recognition memory of early-life polar lipid supplementation using the translational pig model. Twenty-eight intact (i.e., not castrated) male pigs were provided either a control diet (n = 14) or the control diet supplemented with polar lipids (n = 14) from postnatal day 2 until postnatal week 4. After postnatal week 4, all animals were provided the same nutritionally-adequate diets until postnatal week 24. Pigs underwent magnetic resonance imaging at 8 longitudinal time-points to model brain macrostructural and microstructural developmental trajectories. The novel object recognition task was implemented at postnatal weeks 4 and 8 to evaluate recognition memory. Subtle differences were observed between groups in hippocampal absolute brain volumes and fractional anisotropy, and no differences in myelin water fraction developmental patterns were noted. Behavioral outcomes did not differ in recognition memory, and only minimal differences were observed in exploratory behaviors. Our findings suggest that early-life dietary supplementation of polar lipids has limited effect on brain developmental patterns, object recognition memory, and exploratory behaviors.

## 1. Introduction

Breast milk is the optimal early-life nutrition for infants as it contains all of the necessary nutrients that provide the infant protection against infection and disease [1] and support infant growth and brain development [2]. In situations when human milk is not available, infant formula is the only alternative as it contains a nutrient profile similar to that of breast milk. Infant formulas are regularly improved by incorporating new nutrients in the effort to mimic breast milk composition and to aid formula fed infants in having similar overall growth and development to that of breastfed infants. The incorporation of milk polar lipids in formula is gaining interest due to their presence in breast milk [3], their bioactive properties and ability to support human health [4,5], and their potential role in brain development [6]. Examples of polar lipids include glycolipids, sphingolipids, and phospholipids such as phosphatidylcholine, phosphatidylethanolamine, phosphatidylinositol, and phosphatidylserine [7]. These amphiphilic compounds have a variety of functions, some of which include determining permeability of water and nanoparticles through membranes [8], membrane structure and fluidity [9], cell signaling [10,11], and proliferation and differentiation of microglia [12]. Due to their broad functionality and suggested roles in supporting brain development, polar lipids are promising candidates for incorporation into infant formulas.

Indeed, several investigations have evaluated the ability for dietary polar lipids to support neurodevelopment and memory function. In milk, polar lipids are primarily located in milk fat globule membrane (MFGM), a complex triple layer membrane in human milk that has been known to support brain development and cognition [13]. In fact, infants receiving formula supplemented with MFGM performed better on the Bayley Scales of Infant and Toddler Development than those provided with a standard infant formula [14]. Additionally, growth-restricted rats supplemented with MFGM had better T-maze behavioral performance and exhibited early up-regulation of genes involved in brain function compared with growth-restricted rats provided the control diet [15]. Infants that were provided a formula supplemented with gangliosides using complex milk lipids experienced increased intelligence quotient (IQ) scores associated with hand and eye coordination, performance, and overall IQ [16]. Supplementation of individual polar lipids has also been observed to provide positive health impacts. Phosphatidylcholine is especially known to be essential for the neonatal brain as choline supplementation not only supports the early fetal development, but postnatal supplementation of choline also enhances and supports memory formation [17,18]. In addition, dietary supplementation of sphingomyelin may be critical in supporting brain development. When developing rats had experimental inhibition of sphingolipid biosynthesis, dietary sphingomyelin was able to contribute to myelination within the central nervous system [19]. In humans, an association was observed between breastmilk sphingomyelin levels and myelination as well as cognition assessed by the Bayley Scales of Infant and Toddler Development [20]. Research has only brushed the surface of the neurological and neurodevelopmental effects from dietary polar lipid supplementation on postnatal development. Thus, more investigations on these relationships are warranted.

The pig is an accepted preclinical model within the scientific community due to the many anatomical, physiological, and metabolic similarities between humans and pigs. Humans and pigs have a comparable digestive physiology [21]. The eating patterns and nutrient requirements of the species resemble one another during the periods of infancy, growth, reproduction, lactation, and adulthood [22,23]. Additionally, the pig is an increasingly popular animal for neuroscience related research as they possess more resemblances in brain anatomy and growth to humans than many of the commonly used small animal models [24]. Unlike the rodent brain that is smooth, human and pig brains have gyri and sulci and experience a growth spurt within the brain perinatally rather than prenatally or postnatally [25,26,27]. Therefore, we sought to use the pig as a translational animal model to evaluate the longitudinal influence of early-life dietary polar lipid supplementation on neurodevelopment and behavior. Specifically, the dietary intervention consisted of a mixture of sphingomyelin, phosphatidylcholine, phosphatidylethanolamine, and phosphatidylinositol. Dietary supplementation of these compounds individually and together within MFGM are known to support health and development. Therefore, we wanted to explore the developmental and behavioral outcomes from a combination of polar lipids that are not within MFGM. We hypothesized that consumption of dietary polar lipid-supplemented milk replacer would elicit longitudinal improvements in structural and functional aspects of brain development in the young pig and may influence object recognition and exploratory behaviors.

## 2. Materials and Methods

### 2.1. Animals and Housing

All animals and experimental procedures were conducted in accordance with the National Research Council Guide for the Care and Use of Laboratory Animals and approved by the University of Illinois at Urbana-Champaign Institutional Animal Care and Use Committee. Pregnant sows were obtained from a commercial swine farm (Carthage Veterinary Services, Carthage, IL, USA) and transferred to the Veterinary Medicine Research Farm located at the University of Illinois at Urbana-Champaign one week prior to the farrowing (i.e., pig-specific term for giving birth). Sows were provided 15 mg of Matrix (Merck Animal Health/Intervet Inc, Madison, NJ, USA) daily per os for 2–4 d prior to the expected farrowing date and injected with 0.7 mL of Estrumate (250 µg/mL; Merck Animal Health/Intervet Inc, Madison, NJ, USA) one day before the expected farrowing date, to synchronize farrowing events between sows; this is a standard agricultural practice to coordinate farrowing across a contemporary group of sows. Once sows farrowed, pigs were provided with a single dose of prophylactic antibiotic (5.0 mg/kg body weight; Excede, Zoetis, Kalamazoo, MI, USA) and iron dextran (200 mg per pig; Uniferon 200, Pharmacosmos, Watchung, NJ, USA) within the first 24 h of birth. 

Pigs (n = 28) were allowed access to colostrum for up to 48 h to more closely mimic human conditions as most formula fed infants receive the nutrient and immune factor-rich colostrum before formula feeding [28]. Animals were then transferred to the Piglet Nutrition and Cognition Laboratory and artificially reared through postnatal day (PND) 30. Upon arrival at the facility, pigs received two doses of *Clostridium perfringens* antitoxin C and D (one 5 mL dose given subcutaneously and one 3 mL dose given orally; Colorado Serum Company, Denver, CO, USA) as a prophylactic measure on PND 2 to avoid incidence of enterotoxemia that sometimes occurs in young pigs. After PND 30, pigs were transferred to the Veterinary Medicine Research Farm and were group-housed until postnatal week (PNW) 24, i.e., the approximate age of sexual maturity for pigs [29].

The individual custom rearing units (87.6 cm long, 88.9 cm wide, 50.8 cm high) within the Piglet Nutrition and Cognition Laboratory were composed of three acrylic walls, one stainless steel wall and vinyl-coated, expanded-metal flooring. Pigs were allowed to see, hear, and smell, but not touch, neighboring pigs in these units. Each pig was provided a toy for enrichment in their home-cage and were allowed to physically interact with one another for approximately 15 min each day. Lights were automatically controlled (12-h cycle, on from 0800 h to 2000 h) with the ambient temperature set at 26.6 °C for the first 21 d of the study and gradually lowered to 22 °C during the last seven days of housing at the Piglet Nutrition and Cognition Lab. As pigs are typically weaned after 4 weeks-of-age, from PND 30—PNW 8 pigs were transferred back to the Veterinary Medicine Research farm and group-housed in age-appropriate, raised deck pens (1.219 m × 1.219 m; vinyl-coated, expanded metal flooring, with 1 nipple drinker and 4-hole feeder per pen; ad libitum access to feed). After PNW 8, all pigs were moved to larger pens (1.676 m × 3.658 m; solid concrete floors with 1 nipple drinker and feed provided twice-daily per pen (feed amount based on pig age and pen group weight) through study conclusion at PNW 24.

### 2.2. Dietary Groups

Pigs were allotted based on initial body weight to one of two custom bovine milk-based milk replacer treatments (TestDiet, Richmond, IN, USA) formulated to meet all nutritional requirements for the young pig [30]. The control diet (CONT; n = 14) was formulated with whey protein isolate that did not contain polar lipids while the test diet (TEST; n = 14) was formulated with a uniquely processed whey protein source enriched in alpha-lactalbumin containing relatively high levels of sphingomyelin (475 ± 59 mg/100 g), phosphatidylcholine (432 ± 47 mg/100 g), phosphatidylethanolamine (564 ± 143 mg/100 g), and phosphatidylinositol (190 ± 37 mg/100 g). The two diets were aimed at being isocaloric (CONT 4.26 kcal/kg and TEST 4.29 kcal/kg). Milk replacer was reconstituted fresh daily at 200 g of dry powder per 800 g of tap water. Pigs were fed ad libitum using an automated milk replacer delivery system that dispensed milk from 1000 h to 0600 h the next day. Daily records of leftover milk from the previous day and individual pig body weights were documented. The remaining volume of milk was subtracted from the initial volume provided to quantify milk disappearance over the 20-h feeding period, which will henceforth be referred to as milk intake. Once all animals were transitioned to group-housing at the Veterinary Medicine Research Farm after PNW 4, all pigs were maintained on a common series of industry-standard, nutritionally-adequate diets through PNW 24 (i.e., study conclusion). Pigs were transitioned from the milk replacer treatments to the industry-standard diets at PNW 4 as this is the typical age of weaning for domesticated pigs. Individuals involved in conducting and analyzing study results remained blinded to the identity of the early-life dietary treatments until final analyses had been completed.

### 2.3. Imaging Acquisition

At 1, 2, 3, 4, 8, 12, 18, and 24 weeks of age, pigs underwent magnetic resonance imaging (MRI) procedures at the Beckman Institute for Advanced Science and Technology (University of Illinois, Urbana, IL, USA) Biomedical Imaging Center using a MAGNETOM Prisma 3T MRI scanner (Siemens, Munich, Germany). A custom 8-channel head coil designed for young pigs was used through PNW 4 (Rapid Biomedical, Rimpar, Germany) and a 32-channel spine and 18-channel flex coils (Siemens, Munich, Germany) were used for scans occurring from PNW 8–24. Upon arrival to the imaging facility, pigs anesthetized using a combination of telazol:ketamine:xylazine solution (50.0 mg tiletamine plus 50.0 mg of zolazepam reconstituted with 2.50 mL ketamine (100 g/L) and 2.50 mL xylazine (100 g/L); Fort Dodge Animal Health, Overland Park, KS, USA) by i.m. injection at 0.03 mL/kg of body weight. Once anesthetized, pigs were places in a supine position in the MRI machine and kept under sedation by inhalation of isoflurane (0.6 to 2.0% using a progressive dosing regimen based on body weight) with the balance as pure oxygen throughout the entire procedure (total scan time was approximately 75 min per session). Oxygen saturation levels and heart rate were monitored using two pulse oximeters (LifeWindow LW9x, Boynton Beach, FL, USA and MEDRAD Veris 8600, Indianola, PA, USA) each with an infrared sensor that was clipped on the pig’s tail and/or left hind hoof. Observational records of heart rate, partial pressure of oxygen, and percent of isoflurane were recorded every 5 min after anesthetic induction. The pig neuroimaging protocol included a magnetization prepared rapid gradient-echo sequence and diffusion tensor imaging to assess brain macrostructure and microstructure, respectively. The multicomponent driven equilibrium single pulse observation of T_1_ and T_2_ technique was used to measure myelin-associated water fraction (MWF). Imaging techniques are similar to those discussed previously [31] and are described in greater detail below.

A T1-weighted a magnetization prepared rapid gradient-echo sequence was used to obtain anatomic images of the pig brain throughout the 24 wk study. The following sequence-specific parameters were used to acquire T1-weighted a magnetization prepared rapid gradient-echo data through PNW 4: repetition time = 2000.0 ms; echo time = 2.05 ms; inversion time = 1060 ms, flip angle = 9°, matrix = 288; slice thickness = 0.6 mm. Parameters for pigs from PNW 8–24 were as follows: repetition time = 2060.0 ms; echo time = 1.71 ms; inversion time = 1060 ms, flip angle = 9°, matrix = 256; slice thickness = 1.0 mm The final voxel size was 0.6 mm isotropic across the entire head from the tip of the snout to the cervical/thoracic spinal cord junction in pigs through PNW 4 and was 1.0 mm isotropic in pigs from PNW 8–24. Detailed image processing and volume estimation methods has been previously described [32].

Diffusion tensor imaging was used to assess white matter maturation and axonal tract integrity using a diffusion-weighted echo planar imaging sequence. The following parameters were utilized for pigs aged 4 weeks and younger: repetition time = 5100 ms; echo time = 70 ms; generalized auto-calibrating partially parallel acquisitions accelerated by a factor of 2 in the phase encode direction; diffusion weightings = 1000 and 2000 s/mm^2^ across 30 directions; 1 image with a b-value of 0 s/mm^2^. Fifty slices with a 1.6 mm thickness were collected with a matrix size of 100 × 100 for a final voxel size of 1.6 mm isotropic. In pigs older than 4 weeks, the parameters were as follows: repetition time = 5600 ms; echo time = 70 ms; generalized auto-calibrating partially parallel acquisitions accelerated by a factor of 2 in the phase encode direction; diffusion weightings = 1000 and 2000 s/mm^2^ across 30 directions; 1 image with a b-value of 0 s/mm^2^. Fifty-four slices with a 2.0 mm thickness was collected with a matrix size of 130 × 130 for a final voxel size of 2.0 mm isotropic. Diffusion-weighted echo planar imaging images were assessed in the FMRIB Software library [33] to generate values of fractional anisotropy (FA) using methods previously described [32]. Atlas-generated white matter indicated the use of white matter prior probability maps from the pig brain atlas that were used as a region of interest mask. Likewise, DTI-generated white matter indicated a threshold of 0.2 was applied to FA values, thus restricting analysis to white matter tracts only.

Brain myelination patterns were elucidated by measuring MWF utilizing the multicomponent driven equilibrium single pulse observation of T_1_ and T_2_ technique. For pigs 4 weeks and younger, a constant 7.0 × 10.8 × 14.7 mm^3^ sagittally oriented field of view with 160 × 160 × 125 imaging matrix was used, providing a voxel volume of 1.7 × 1.7 × 2.6 mm^3^. The spoiled gradient echo and T_2_/T_1_—weighted balanced steady-state free precession (SSFP) data were acquired with the following sequence-specific parameters: spoiled gradient echo, echo time (TE)/repetition time (TR) = 2.7 ms/5.6 ms; receiver bandwidth = 350 Hz/voxel; and SSFP, TE/TR flip angles = 2.6 ms/5.3 ms; receiver bandwidth = 350 Hz/voxel. For pigs older than 4 weeks, a constant 4.1 × 5.4 × 31.5 mm^3^ sagittally oriented field of view with 260 × 260 × 240 imaging matrix was used, providing a voxel volume of 2.7 × 2.7 × 3.0 mm^3^. The spoiled gradient echo and SSFP data were acquired with the following sequence-specific parameters: spoiled gradient echo, TE/TR = 2.7 ms/5.6 ms; receiver bandwidth = 350 Hz/voxel; and SSFP, TE/TR = 2.6 ms/5.3 ms; receiver bandwidth = 350 Hz/voxel. Two sets of SSFP data were acquired with phase-cycling increments of both 0° and 180° to allow for correction of main magnetic field (i.e., off-resonance) artifacts. Processing of MWF data was performed using methods described previously [34] with modifications to the sequence only including a change in the threshold used for imaging data collected for the pig.

### 2.4. Behavioral Testing

Novel object recognition (NOR) is a behavioral paradigm used to assess recognition memory as a primary indicator of cognitive behavior of the pig, as described in detail previously [31,35]. Testing consisted of a habituation phase, a sample phase, and a test phase, with a delay period between the sample and test phases. During the habituation phase, each pig was placed in an empty testing arena for 10 min for two days leading up to the sample phase. In the sample phase, pigs were given 5 min in the arena to explore two identical objects. After a delay of 48 h, the test phase was conducted for 5 min where pigs were returned to the arena which contained one object from the sample phase as well as a novel object. Between trials, objects were removed, immersed in hot water with detergent, and rubbed with a towel while the arena was sprayed with water to remove any odor cues from urine and feces of previous pig. Objects chosen had a range of characteristics (i.e., color, texture, shape, and size); however, the novel and sample objects only differed in shape and size. Only objects previously shown to elicit a null preference were used for testing. The NOR task was completed at two different time-points, PNW 4 and 8. When pigs were 4 weeks-of-age, the habituation trial began at PND 24 and the testing trial at PND 28. When pigs were 8 weeks-of-age, habituation trials began at PND 52 and the testing trial began at PND 56. The object set used for each time-point was different and counterbalanced per pig, and all pigs that completed the study were tested on the NOR task. Recognition index, the proportion of time spent investigating the novel object compared with the total exploration time of both objects, was compared to a chance performance value of 0.50 to assess recognition memory. Values greater than 0.50 were interpreted as being indicative of a novelty preference, thus suggesting pigs exhibited recognition memory.

### 2.5. Statistical Analysis

All statistical models employed included replicate and litter as a random effect. The level of significance was set at *p* < 0.05. Growth and milk intake data for each individual pig were subjected to an analysis of variance using SAS (version 9.3; SAS Inst. Inc., Cary, NC, USA). Data for growth and milk intake were analyzed as a repeated-measures analysis of variance also using the MIXED procedure, and exploratory behavior was analyzed as a repeated measures analysis of variance with diet as main effect and age as repeating factor. To assess recognition memory, the recognition index was compared to a chance performance value of 0.50 using a one-sample t-test.

Absolute volume, FA, and MWF developmental patterns were constructed for each brain area in each pig using the NLMIXED method in SAS 9.3. Parameter estimations were computed for nine different sigmoid-type models. The growth models of Gompertz [36], Bleasdale and Nelder [37], Richards [38], Stannard [39], a modified Gompertz from Dean III et al. [40], two different logistic functions [40,41], the generalized logistic, and the hyperbolic tangent were all fitted. The Bayesian information criterion (BIC) was measured for each model in each ROI. The BIC is a criterion for model selection where the lowest BIC is preferred because unexplained variation in the dependent variable, and the number of explanatory variables increase the value of BIC [42]. The Gompertz was chosen as the best model to use for absolute volumes, FA, and MWF data because it had the best Bayesian information criterion value, a criterion for model selection, across all brain regions using the highest ranked sum for those specific data, as discussed previously [31].

The Gompertz model was parameterized as follows:(1)Outcome=a∗exp(−1∗exp(b−g∗PNW))
where the outcome was either absolute volume, FA, or MWF, and *PNW* indicated postnatal week (i.e., age of the pig). Parameter estimations for each outcome were computed for maximum absolute (i.e., plateau) value (*a*), onset of initial developmental increase (*b*), and overall rate of development (*g*). A two-sample t-test was conducted using SAS version 9.3; SAS Inst. Inc., Cary, NC, USA) to compare individual modeled parameter estimates between CONT and TEST pigs.

## 3. Results

### 3.1. Growth and Intake

#### 3.1.1. Body Weight

Pig body weights did not differ (*p* = 0.876) between dietary treatments, though the main effect of PNW was observed for daily body weights (*p* < 0.001), meaning that pigs grew throughout the study period (Figure 1). The lack of an interaction effect on body weight (*p* = 0.687) signified that this outcome was not influenced by the combination of PNW and the early-life dietary intervention, keeping in mind that all pigs received a common series of diets starting at PND 30.

#### 3.1.2. Feed Intake

A main effect of PND (*p* < 0.001) was observed, meaning that pigs consumed progressive amounts of feed over time. A main effect of dietary treatment on voluntary intake was not observed (*p* = 0.734), and the lack of interaction on intake (*p* = 0.958) signified that overall intake was not influenced by dietary treatment over the study duration.

#### 3.1.3. Growth Performance 

Effects of diet on average daily body weight gain, average daily feed intake, and the gain-to-feed ratio (i.e., efficiency of body weight gain) are displayed in Table 1. Average daily gain and gain-to-feed ratio were similar between groups throughout the study duration. Only a main effect of diet was observed for average daily feed intake during PNW 1–4, with TEST pigs consuming 7.7% more (*p* = 0.031) milk replacer solids compared with CONT pigs. Average daily feed intake was comparable between dietary groups once all pigs were receiving the common series of diets and through the duration of the study.

### 3.2. Neuroimaging Outcomes

The developmental patterns of regional and whole brain absolute volumes were similar between CONT and TEST pigs, except for an increased overall rate of development of the right hippocampus in TEST pigs (Figure 2; Table 2). Additionally, CONT and TEST pigs exhibited similar regional FA developmental patterns, but TEST pigs had a greater estimated plateau FA value for the left hippocampus compared with CONT pigs (Table 3). Lastly, no differences due to early-life dietary treatment were detected for MWF developmental patterns (Table 4).

### 3.3. Behavioral Outcomes

Here, the behavioral outcomes were measured on two groups of pigs, receiving CONT and TEST diets, at two different time-points, PNW 4 and 8. Therefore, there were 4 treatment groups in total (CONT pigs at PNW4 and PNW 8, and TEST pigs at PNW 4 and PNW 8). Recognition memory, as assessed using the NOR task, is displayed in Table 5. 

None of the four groups demonstrated a novelty preference with recognition index being different (*p* > 0.05) than the chance level of 0.50. Neither the main effects of diet or age nor their interaction were observed for recognition memory (*p* > 0.05). The main effect of age (*p* = 0.001) and the interaction effect of diet and age (*p* = 0.004) were observed for latency to first object visit, with the TEST pigs at PNW 4 having longer latency than any other treatment groups (Table 6). Similar results were observed for latency measures of novel and sample object interactions, as pigs at PNW 4 had longer latency to the first novel object visit (*p* = 0.006; Table 7). Additionally, there was an interaction effect for latency to first sample object visit, with TEST pigs having longer latency (*p* = 0.024) at PNW 4 compared with other groups (Table 8).

## 4. Discussion

Polar lipids represent integral components of all biological membranes and serve a variety of structural and biological functions within the cell, including in support of intestinal health and development [43] and exhibiting anti-inflammatory properties [43,44,45]. Additionally, research has suggested that early-life polar lipid supplementation can help to narrow the gap in neuronal development and cognitive performance between breastfed and formula-fed infants [14,46]. Most of the previously conducted research examined the developmental benefits of dietary supplementation of only one particular polar lipid, polar lipids within MFGM, or evaluated supplementation effects at a single time-point [14,20,46,47,48]. Thus, we sought to broaden the field by evaluating the longitudinal influence of relatively high levels (compared to pig milk levels) of dietary supplementation with sphingomyelin, phosphatidylcholine, phosphatidylethanolamine, and phosphatidylinositol on body growth, brain development, and cognitive outcomes. Contrary to our hypothesis, our results indicate early-life supplementation with this blend of polar lipids elicited minimal impact on the structure and function, in terms of behavior and brain measures. The only differences between the groups were observed in the overall rate of development in the absolute volume of the right hippocampus and maximum absolute FA value of the left hippocampus.

The rate and efficiency of whole-body growth of pigs in our study was comparable with domestic pigs raised in a commercial setting. TEST pigs had higher average daily feed intake than CONT pigs from PNW 1–4, but this difference did not influence overall growth performance during the 24-week study. Pigs from both groups had comparable body weights throughout the study, indicating that dietary polar lipids did not affect overall growth. Similarly, studies on rodents and human infants examining the health outcomes associated with polar lipid supplementation from MFGM have observed no weight differences between groups receiving an MFGM-supplemented formula versus an non-supplemented control [20,46,49]. Moreover, preterm infants provided milk fortified with sphingomyelin did not have body weight differences compared with preterm infants provided milk with lower sphingomyelin content [47]. This indicates that body weight is not likely impacted by the consumption of dietary polar lipids. 

In our pig model where early-life polar lipid supplementation was evaluated for its effects on longitudinal outcomes, brain development was evaluated by estimating absolute volume, FA, and MWF developmental trajectories. Volumetric analyses were conducted on whole brain, grey matter, white matter, cerebrospinal fluid, and twenty-four sub-regions. Only an increased overall rate of development of the right hippocampus was observed in TEST pigs in contrast to CONT pigs, but no other differences were observed in any region measured, including whole brain (Figure 2). This is in contrast to a previous study that found higher volumes and more grey and white matter in pigs fed a diet supplemented with phospholipids and gangliosides [50]. However, the MRI measures in the previous study were only evaluated at a single time-point, PNW 4, rather than evaluating longitudinal outcomes. Nevertheless, the maximum absolute volume estimates for whole brain and all other sub-regions were similar for TEST and CONT pigs, signifying that polar lipid supplementation had little influence on longitudinal absolute volume development.

Diffusion tensor imaging and myelin water imaging were utilized to measure brain microstructural changes from FA and MWF data, respectively. Specifically, FA provides insight into fiber integrity by reflecting the degree of water diffusion anisotropy [51,52] while MWF serves to detect myelin content and changes in myelination within the brain [53,54]. Polar lipids are key structural components of cell membranes and neural tissues [4,5] and have been suggested to promote myelination [17]. Thus, we expected to observe brain microstructural differences in terms of myelination between pigs that consumed diets low (CONT) or high (TEST) in polar lipid supplementation during early-life. However, only minor differences were noted between early-life dietary treatment groups. Most estimates for the FA developmental pattern were similar between CONT and TEST pigs, with the exception of the left hippocampus having a higher estimated plateau FA value for TEST pigs as compared with CONT pigs. Moreover, the developmental patterns of MWF were comparable between CONT and TEST pigs for each region of interest measured. This suggests that early-life dietary polar lipid consumption influenced the microstructure of the left hippocampus, but this was independent of changes in myelin development. Potential microstructural differences may have included changes in intra-axonal organization, synaptogenesis, fiber diameter, and fiber density [52]. Utilizing immunohistochemistry, Guillermo et al. (2015) similarly found that rats given a diet supplemented with complex milk lipids (i.e., enriched with phospholipids and gangliosides) did not experience a difference in myelination as compared with rats given a non-supplemented diet. However, they observed broader areas of synaptophysin, elevated astrocytes in the hippocampus, and altered dopamine neuroplasticity in the striatum of rats supplemented with complex milk lipids. They concluded that although the complex milk lipids did not alter myelination, the lipids did moderately promote synaptic plasticity. Therefore, the impact of polar lipid supplementation may be occurring on a microscopic level that can only be elucidated through histological or immunohistochemistry analyses and not necessarily through more gross measures obtained using neuroimaging techniques. 

Effects of dietary polar lipid supplementation during early life were studied not only on brain structural development but also on behavioral outcomes. The NOR task is a well-established behavioral paradigm for young pigs, which has been utilized in various nutritional neuroscience studies to assess recognition memory and exploratory behaviors [35]. From the current study, no differences due to diet or age were observed for recognition memory or most exploratory behaviors expressed by pigs. However, outcomes from the latency measures from NOR revealed that TEST pigs had longer latency to the first object visit at PNW 4, which is congruent with differences in latency measures for interaction with the sample object. Previously, a relationship between NOR task latency measures and anxiety-related exploratory behaviors has been speculated [55], suggesting that early-life dietary supplementation of polar lipids may modulate anxiety-related exploratory behaviors. However, it is important to note that differences in exploratory behaviors driven during the NOR task were of small magnitude in the current study. Therefore, further research is necessary to elucidate the influence of early-life dietary polar lipid supplementation on memory and exploratory behaviors related to emotionality.

Here, we demonstrated that early-life dietary supplementation of polar lipids had no detrimental influence on the development of brain morphology and only a moderate impact on anxiety-related exploratory behavior in a recognition memory task. These were interesting findings as other investigations have observed that polar lipid supplementation in infant formula supported brain development and cognition. For example, lipid extracts from MFGM demonstrated improved IQ scores in human infants [14]. Similarly, young pigs provided a supplementation of dietary phospholipids and gangliosides exhibited an improvement in spatial T-maze performance [50] compared with pigs consuming a non-supplemented diet. Additionally, an increase in novelty recognition was observed in rodents supplemented with complex lipids [56] compared with non-supplemented rodents. Moreover, MFGM and prebiotics supplementation during early-life helped attenuate anxiety-related behaviors in rodents [57].The discrepancy in outcomes between the current study and previous research may be due to differences in the structural organization in which the polar lipids are presented within the diet, as this may contribute to their functionality. Indeed, the size and structure of lipids in dairy products determines their bioavailability, digestion, and absorption [58,59,60,61]. Moreover, we used an extracted fraction of the complete MFGM complex that would be found in milk, while other studies utilized the MFGM complex itself or other mixtures of lipids. The different bioactive elements within MFGM and lipid mixtures may have had a synergistic effect that contributed to more developmental outcomes than what was observed with solely polar lipid supplementation. For instance, apart from their basic function as structural components within MFGM, research has found that gangliosides within the brain contribute to a multitude of developmental functions, some of which include the transmission and storage of information [62], integrity of nerve tissue [63], and regulatory roles in axonogenesis [64]. Dietary gangliosides have been observed to alter ganglioside levels within the brain [65] and positively influence brain development [66]. Alternatively, improving nutrition alone may have been insufficient to elicit developmental effects. Undernourished children who received zinc supplementation benefited the most in overall development when also receiving psychosocial stimulation through activities such as language development and problem solving compared with unstimulated children [67]. Thus, more developmental and behavioral outcomes may have been observed had pigs been stimulated with cognitive behavioral tests at more than two time-points. Moreover, limitations in the NOR arena size prevented the ability to test pigs at ages later than PNW 8, in which potential long-term cognitive outcomes may had been exhibited. The current study was conducted longitudinally, beginning shortly after birth and ending at the sexual maturity of the pig. Additionally, pigs received polar lipid supplementation only early in life, from PND 2 to PNW 4. Although there is vast literature evaluating polar lipids and their potential contributions to development, studies typically evaluate developmental outcomes at only one time-point or after a year or two after supplementation. Hence, most previous work may have only observed short-term beneficial outcomes from supplementation. Interestingly, infants provided with bovine MFGM supplemented infant formula that had better cognitive performance than infants provided with a standard diet displayed no long-term cognitive impact from the early-life supplementation at 6.5 years of age [68]. Thus, more longitudinal-based research is warranted to fully evaluate the long-term benefits of early-life polar lipid supplementation. 

The current study is not without its limitations. The pigs utilized in this study were intact males as the animals would be reproductively mature and large by the study conclusion. Therefore, including both sexes would require raising them in separate facilities and having different personnel manage the different sexes, which would increase the variability in the dataset. Neuroimaging and behavioral outcomes may have been influenced by the fatty acid profile of the diets, but analysis of the fatty acid profile for each diet (early-life TEST, early-life CONT, and the industry-standard, nutritionally-adequate diet) was not conducted. For neuroimaging measures, two different MRI coils and sequences had to be utilized to accommodate the massive differences in head size of pigs from early ages to older ages. Thus, the spatial resolution differences in the imaging sequence may have attributed to variability within the dataset. Additionally, there was higher variability and therefore higher standard errors in the later age time-points of the pigs due to larger pigs having heavier and deeper breathing that contributed to movement in the MRI scanner. The NOR task successfully measures recognition memory during early-life in pigs; however, the addition of other behavioral tests investigating other measurements of cognitive development such as radial arm maze or social recognition task was not plausible due to the restrictions in number of pigs and the timeline of the study design.

## 5. Conclusions

This study examined the longitudinal outcomes of early-life dietary polar lipid supplementation on neurodevelopment and recognition memory in the domestic pig. The intervention study did not result in any harmful effects on pigs’ physical growth, brain development or object recognition memory and exploratory behaviors. Minor differences were observed between CONT and TEST pigs in growth performance throughout the study. Moreover, the hippocampus exhibited a higher rate of development in absolute brain and a higher maximum FA value in TEST pigs than CONT pigs, but no differences were recorded on MWF developmental pattern. No differences in recognition memory were observed between CONT and TEST pigs, and only minimal differences in exploratory behaviors were observed, with the exception of latency measures. Overall, dietary polar lipid supplementation during early-life resulted in a small impact on hippocampus maturation and otherwise minimal differences in brain, recognition memory, and exploratory behaviors over a 24-week study period in pigs.

## Figures and Tables

**Figure 1 nutrients-13-02486-f001:**
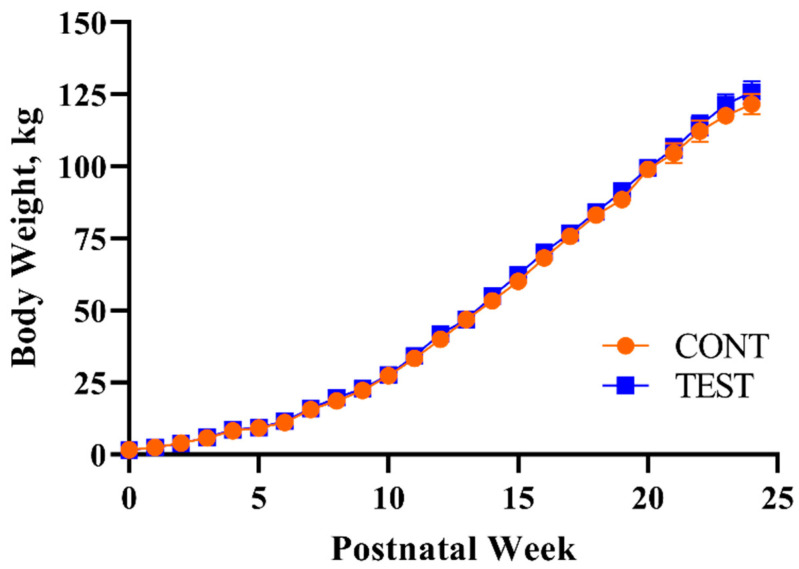
Weekly body weights of pigs receiving early-life diets differing in polar lipid composition. Abbreviations: CONT, control diet; TEST, polar lipid-supplemented diet.

**Figure 2 nutrients-13-02486-f002:**
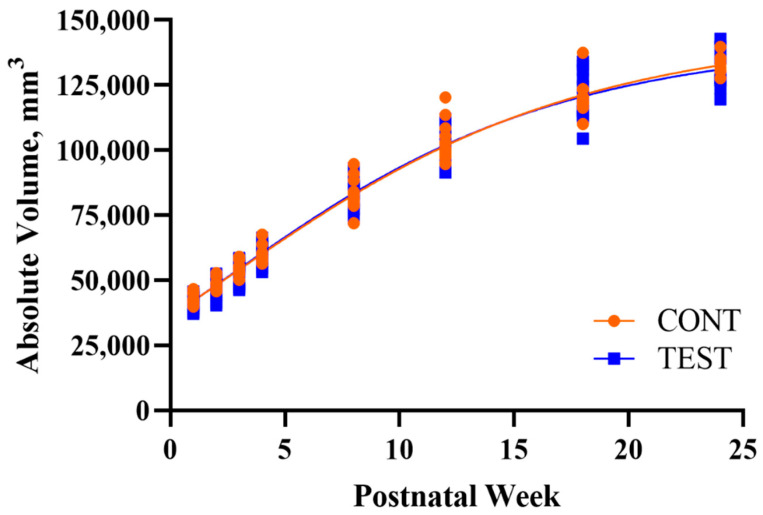
Developmental patterns of whole brain absolute volume in pigs receiving early-life diets differing in polar lipid composition. Abbreviations: CONT, control diet; TEST, polar lipid-supplemented diet.

**Table 1 nutrients-13-02486-t001:** Growth and feeding performance on milk replacer (PNW 1–4) and feed (PNW 5–24) ^1^.

	Diet		Pooled	
Measure	CONT	TEST	SEM	*p*-Value ^2^
PNW 1–4				
ADG, kg/day	0.245	0.260	0.0108	0.233
ADFI, kg solids/day	0.260	0.280	0.0801	0.031
G:F, kg BW:kg solids	0.188	0.184	0.0049	0.556
PNW 5–24				
ADG, kg/day	0.818	0.845	0.0361	0.355
ADFI, kg solids/day	1.904	2.000	0.1054	0.431
G:F, kg BW:kg solids	0.483	0.424	0.0637	0.455

^1^ Abbreviations: ADG, average daily gain; ADFI, average daily feed intake; BW, body weight; CONT, control diet; G:F, gain to feed; PND, postnatal day; PNW, postnatal week; SEM, standard error of the mean; TEST, test diet. ^2^ *p*-values derived from a repeated-measures mixed model ANOVA comparing pigs provided different early-life dietary interventions.

**Table 2 nutrients-13-02486-t002:** Brain region-specific parameter estimates for absolute volumes generated using the Gompertz model ^1^.

	CONT			TEST			Pooled SEM			*p*-Value ^2^		
ROI Parameter ^3^	a	b	g	a	b	g	a	b	g	a	b	g
Whole brain	147,805.08	0.34	0.11	151,390.00	0.33	0.11	7221.096	0.021	0.010	0.634	0.722	0.987
Grey matter	60,903.83	0.20	0.11	62,357.33	−0.07	0.13	5604.781	0.210	0.041	0.798	0.180	0.680
White matter	58,858.29	0.56	0.13	58,095.33	0.59	0.13	3168.560	0.027	0.010	0.812	0.329	0.515
Cerebrospinal fluid	29,378.14	1.13	0.17	26,629.08	1.09	0.17	2519.084	0.052	0.021	0.264	0.486	0.904
Cerebral aqueduct	82.46	0.04	0.09	64.92	−0.14	0.15	14.475	0.173	0.080	0.238	0.321	0.488
Corpus callosum	1500.20	0.38	0.08	1711.90	0.47	0.07	391.500	0.081	0.018	0.594	0.265	0.707
Cerebellum	24,387.70	0.42	0.08	34,678.80	0.56	0.08	9305.100	0.107	0.018	0.281	0.215	0.874
Fourth ventricle	227.80	−0.05	0.06	187.80	−0.19	0.07	65.106	0.234	0.011	0.546	0.557	0.355
Hypothalamus	388.70	0.36	0.09	400.80	0.36	0.09	49.367	0.052	0.011	0.809	0.950	0.908
Left caudate	1102.40	0.32	0.06	1235.90	0.29	0.07	277.300	0.122	0.016	0.646	0.824	0.563
Left hippocampus	1388.50	0.41	0.08	1718.10	0.49	0.07	332.000	0.087	0.017	0.331	0.350	0.552
Left inferior colliculus	435.00	0.58	0.05	389.00	0.46	0.06	71.070	0.129	0.015	0.525	0.369	0.477
Left internal capsule	2021.30	0.26	0.08	2759.70	0.43	0.06	395.000	0.102	0.010	0.107	0.129	0.095
Left olfactory bulb	3432.50	0.36	0.09	3567.80	0.42	0.11	459.800	0.151	0.018	0.773	0.700	0.509
Left putamen globus pallidus	831.80	0.32	0.06	845.60	0.43	0.04	229.800	0.147	0.012	0.952	0.473	0.302
Left superior colliculi	1085.80	0.54	0.06	863.80	0.45	0.06	196.400	0.096	0.011	0.270	0.327	0.685
Lateral ventricle	2068.70	0.36	0.06	2330.30	0.43	0.06	439.900	0.101	0.019	0.558	0.471	0.972
Medulla	5210.30	0.60	0.08	8241.60	0.67	0.08	1995.800	0.149	0.018	0.d180	0.665	0.852
Midbrain	6179.30	0.33	0.09	6044.50	0.35	0.09	319.300	0.053	0.011	0.677	0.710	0.642
Pons	4237.10	0.53	0.08	3985.80	0.51	0.08	354.700	0.149	0.015	0.487	0.891	0.928
Right caudate	1256.10	0.39	0.06	2094.30	0.49	0.06	620.000	0.136	0.021	0.216	0.461	0.957
Right hippocampus	1152.20	0.37	0.08	1863.40	0.72	0.05	380.000	0.156	0.011	0.135	0.053	0.009
Right inferior colliculus	489.60	0.60	0.05	524.60	0.53	0.05	91.486	0.127	0.018	0.706	0.552	0.722
Right internal capsule	2134.60	0.34	0.07	2335.30	0.35	0.07	294.700	0.069	0.012	0.503	0.797	0.603
Right olfactory bulb	3687.60	0.32	0.20	3491.80	0.33	0.08	715.400	0.135	0.101	0.781	0.941	0.268
Right putamen globus pallidus	576.00	0.29	0.05	657.40	0.31	0.06	90.235	0.094	0.012	0.396	0.817	0.486
Right superior colliculi	811.10	0.41	0.07	1002.30	0.47	0.07	184.600	0.096	0.018	0.311	0.544	0.716
Thalamus	3563.40	0.36	0.06	3639.10	0.41	0.05	381.100	0.194	0.011	0.845	0.801	0.640

^1^ Abbreviations: CONT, control diet, ROI, region of interest; SEM, standard error of the mean; TEST, test diet. ^2^ *p*-values derived from a two-sample t-test comparing pigs provided different early-life dietary interventions. ^3^ Parameter estimations for each outcome were computed for maximum absolute (i.e., plateau) value (a), onset of initial developmental increase (b), and overall rate of development (g).

**Table 3 nutrients-13-02486-t003:** Brain region-specific parameter estimates for fractional anisotropy generated using the Gompertz model ^1^.

	CONT			TEST			Pooled SEM			*p*-Value ^2^		
ROI Parameter ^3^	a	b	g	a	b	g	a	b	g	a	b	g
Corpus callosum	0.33	−1.01	1.02	0.33	−1.01	1.01	0.006	0.005	0.007	0.554	0.105	0.128
Cerebellum	0.31	−1.00	1.00	0.31	−1.00	1.00	0.002	0.002	0.002	0.585	0.514	0.546
Left caudate	0.37	−1.04	1.00	0.37	−1.05	1.04	0.004	0.044	0.043	0.713	0.807	0.361
Left hippocampus	0.34	−1.01	1.01	0.35	−1.01	1.00	0.005	0.005	0.007	0.014	0.271	0.337
Left internal capsule	0.43	−1.46	1.67	0.43	−1.60	1.90	0.005	0.104	0.151	0.454	0.169	0.137
Left side	0.36	−1.10	1.12	0.36	−1.16	1.17	0.004	0.057	0.066	0.591	0.270	0.449
Right caudate	0.37	−1.01	0.98	0.36	−1.04	1.00	0.006	0.041	0.026	0.283	0.409	0.562
Right hippocampus	0.34	−1.03	1.04	0.35	−1.03	1.02	0.006	0.021	0.026	0.287	0.822	0.471
Right internal capsule	0.43	−1.49	1.73	0.43	−1.63	1.96	0.006	0.098	0.137	0.965	0.171	0.114
Right side	0.36	−1.10	1.09	0.36	−1.15	1.10	0.005	0.067	0.061	0.891	0.468	0.855
Thalamus	0.37	−0.99	0.95	0.38	−1.00	0.98	0.004	0.012	0.021	0.100	0.155	0.170
FA mask	0.34	−1.00	1.00	0.34	−1.00	1.00	0.003	0.001	0.001	0.861	0.530	0.442
White matter mask	0.36	−1.15	1.19	0.36	−1.17	1.18	0.005	0.049	0.058	0.746	0.818	0.917

^1^ Abbreviations: CONT, control diet; FA, fractional anisotropy; ROI, region of interest; SEM, standard error of the mean; TEST, test diet. ^2^ *p*-values derived from a two-sample t-test comparing pigs provided different early-life dietary interventions. ^3^ Parameter estimations for each outcome were computed for maximum absolute (i.e., plateau) value (a), onset of initial developmental increase (b), and overall rate of development (g).

**Table 4 nutrients-13-02486-t004:** Brain region-specific parameter estimates for myelin water fraction generated using the Gompertz model ^1^.

	CONT			TEST			Pooled SEM			*p*-Value ^2^		
ROI Parameter ^3^	a	b	g	a	b	g	a	b	g	a	b	g
Corpus callosum	0.26	−0.50	0.24	0.25	−0.17	0.63	0.113	0.337	0.258	0.970	0.381	0.262
Cerebellum	0.16	1.64	4.01	6.36	15.64	2.46	4.484	13.335	1.352	0.321	0.460	0.368
Combined cortex	0.08	−1.22	0.36	0.25	−0.90	0.66	0.118	0.274	0.265	0.336	0.367	0.372
Combined hippocampus	0.12	−0.71	0.33	0.10	−0.70	0.29	0.032	0.142	0.066	0.581	0.944	0.659
Combined internal capsule	0.24	−0.83	0.18	0.21	−0.41	0.36	0.108	0.291	0.111	0.728	0.154	0.133
Hypothalamus	0.16	1.35	2.64	4.74	−0.24	0.39	3.194	1.516	1.612	0.327	0.445	0.297
Left cortex	0.10	−1.47	0.23	0.26	0.42	1.66	0.126	0.823	0.821	0.381	0.124	0.221
Left hemisphere	0.12	−0.55	2.08	0.12	−14.14	0.81	0.041	9.913	1.458	0.973	0.315	0.513
Left hippocampus	0.47	−0.49	0.33	0.23	0.07	0.57	0.176	0.317	0.184	0.361	0.223	0.337
Left inferior colliculus	0.47	0.97	4.30	4.89	36.92	3.93	3.569	27.766	1.239	0.352	0.355	0.832
Left internal capsule	0.18	−0.81	0.31	0.17	−0.44	0.45	0.070	0.312	0.209	0.913	0.255	0.516
Left olfactory bulb	0.09	−6.37	8.97	0.12	−3.63	9.75	0.059	7.601	3.627	0.691	0.682	0.881
Left putamen globus-pallidus	0.21	−0.37	0.30	6.28	16.77	0.49	4.497	12.274	0.136	0.330	0.317	0.262
Left superior colliculus	0.23	40.74	0.87	0.17	1.74	2.57	0.103	28.316	1.018	0.689	0.286	0.220
Medulla	0.18	−2.52	9.56	5.80	40.49	6.00	4.162	32.027	1.665	0.331	0.334	0.144
Midbrain	0.16	0.99	2.58	0.10	−0.37	1.49	0.038	1.215	1.296	0.247	0.355	0.482
Pons	0.09	5.99	10.57	6.41	39.20	12.26	3.789	29.250	5.239	0.254	0.359	0.752
Right cortex	0.08	−1.26	0.37	0.05	−8.54	0.36	0.043	5.427	0.059	0.642	0.333	0.910
Right hemisphere	0.09	−1.29	0.49	0.09	−0.96	0.55	0.015	0.203	0.118	0.780	0.178	0.698
Right hippocampus	0.09	−0.80	0.35	0.11	−0.71	0.25	0.024	0.120	0.060	0.533	0.621	0.274
Right inferior colliculus	0.18	1.46	4.96	0.21	0.60	6.21	0.102	3.420	2.973	0.866	0.862	0.771
Right internal capsule	0.47	0.05	0.13	0.24	−0.42	0.33	0.153	0.302	0.120	0.131	0.134	0.104
Right olfactory bulb	0.11	−5.24	7.32	0.11	−1.91	6.96	0.048	4.758	2.221	0.997	0.465	0.900
Right putamen globus-pallidus	0.32	−0.22	0.22	0.39	−0.10	0.23	0.168	0.227	0.089	0.664	0.669	0.954
Right superior colliculus	0.09	−0.26	1.13	0.13	0.02	1.42	0.040	0.475	0.628	0.498	0.403	0.385
Thalamus	0.25	−0.66	0.23	0.11	−0.58	0.33	0.101	0.234	0.083	0.329	0.678	0.389
Whole brain	0.11	1.53	3.57	0.13	−0.92	0.82	0.053	2.120	2.144	0.820	0.425	0.379

^1^ Abbreviations: CONT, control diet; ROI, region of interest; SEM, standard error of the mean; TEST, test diet. ^2^ *p*-values derived from a two-sample t-test comparing pigs provided different early-life dietary interventions. ^3^ Parameter estimations for each outcome were computed for maximum absolute (i.e., plateau) value (a), onset of initial developmental increase (b), and overall rate of development (g).

**Table 5 nutrients-13-02486-t005:** Recognition memory on the NOR task with a 48-h delay ^1^.

Diet	*n*	Mean	SEM	*p*-Value ^2^
Week 4				
CONT	12	0.51	0.083	0.471
TEST	8	0.59	0.077	0.132
Week 8				
CONT	15	0.53	0.068	0.357
TEST	10	0.53	0.087	0.355

^1^ Abbreviation: CONT, control diet; NOR, novel object recognition; SEM, standard error of the mean; TEST, test diet. ^2^
*p*-value derived from one-tailed t-test for a recognition index above 0.50.

**Table 6 nutrients-13-02486-t006:** Exploratory behavior of all objects during the test trial of the NOR task ^1^.

Measurements	RI	Total Object Visit Time, s	Number of All Object Visits, n	Mean Object Visit Time, s/Visit	Latency to First Object Visit, s *	Latency to Last Object Visit, s
Effect of diet						
CONT	0.56	47.6	11.5	3.8	4.7	229.6
TEST	0.53	62.0	11.7	4.6	10.4	257.9
SEM	0.059	11.28	1.50	0.68	2.29	18.46
Effect of age						
Week 4	0.51	59.0	12.0	4.6	10.7	239.6
Week 8	0.56	50.7	11.2	3.8	4.4	247.9
SEM	0.065	10.57	1.21	0.71	2.20	20.18
Interaction means						
CONT:Week 4	0.50	56.0	12.3	4.2	4.0 ^a^	241.7
CONT:Week 8	0.59	39.3	10.8	3.4	5.4 ^a^	217.4
TEST:Week 4	0.53	61.9	11.7	5.0	17.4 ^b^	237.5
TEST:Week 8	0.54	62.1	11.7	4.2	3.4 ^a^	278.4
SEM	0.592	15.36	1.77	1.03	3.28	26.67
*p*-Value ^2^						
Diet	0.860	0.354	0.922	0.406	0.450	0.120
Age	0.546	0.501	0.490	0.396	0.001	0.720
Diet × Age	0.633	0.491	0.465	0.993	0.004	0.165

^a,b^ Superscript letters denote differences between treatment means (*p* < 0.05). ^1^ Abbreviation: CONT, control diet; NOR, novel object recognition; RI, recognition index; SEM, standard error of the mean, TEST; test diet. ^2^ *p*-value derived from repeated-measures analysis of variance for the main effects and the interaction. * *p*-values are driven from log transformation due to a violation of the homogeneity of variance assumption.

**Table 7 nutrients-13-02486-t007:** Exploratory behavior of novel objects during the test trial of the NOR task ^1^.

Measurements	Total Novel Object Visit Time, s	Number of Novel Object Visits, n	Mean Novel Object Visit Time, s/Visit	Latency to First Novel Object Visit, s *	Latency to Last Novel Object Visit, s
Effect of diet					
CONT	24.9	5.8	3.7	11.1	193.9
TEST	37.2	5.7	6.0	26.1	221.4
SEM	7.45	0.82	1.16	7.92	25.28
Effect of age					
Week 4	29.9	5.9	4.7	21.9	200.2
Week 8	32.2	5.6	5.0	15.3	215.0
SEM	7.91	0.71	1.25	8.44	23.97
Interaction means					
CONT:Week 4	29.1	6.2	4.0	13.6	184.8
CONT:Week 8	20.7	5.5	3.5	8.7	202.9
TEST:Week 4	30.7	5.6	5.4	30.3	215.6
TEST:Week 8	43.6	5.8	6.6	21.8	227.1
SEM	11.07	1.03	1.68	11.08	32.47
*p*-value ^2^					
Diet	0.193	0.899	0.073	0.294	0.345
Age	0.818	0.683	0.810	0.006	0.516
Diet × Age	0.291	0.539	0.552	0.308	0.886

^1^ Abbreviation: CONT, control diet; NOR, novel object recognition; SEM, standard error of the mean; TEST, test diet. ^2^
*p*-value derived from repeated-measures analysis of variance for the main effects and the interaction. * *p*-values are driven from log transformation due to a violation of the homogeneity of variance assumption.

**Table 8 nutrients-13-02486-t008:** Exploratory behavior of sample object during the test trial of the NOR task ^1^.

Measurements	Total Sample Object Visit Time, s	Number of Sample Object Visits, n	Mean Sample Object Visit Time, s/visit	Latency to First Sample Object Visit, s ^3^	Latency to Last Sample Object Visit, s
Effect of diet					
CONT	21.1	5.4	3.6	11.1	212.3
TEST	25.3	6.5	3.7	20.9	235.4
SEM	5.80	0.89	0.61	5.93	24.67
Effect of age					
Week 4	26.3	6.3	4.2	22.8	223.9
Week 8	20.1	5.5	3.1	9.3	223.8
SEM	5.33	0.97	0.59	5.43	25.74
Interaction means					
CONT:Week 4	26.9	6.5	4.1	12.0 ^ab^	231.3
CONT:Week 8	15.4	4.2	3.1	10.3 ^ab^	193.3
TEST:Week 4	25.8	6.1	4.3	33.6 ^a^	216.4
TEST:Week 8	24.9	6.9	3.2	8.3 ^b^	254.3
SEM	7.75	1.31	0.81	8.15	32.07
*p*-Value ^2^					
Diet	0.599	0.266	0.888	0.415	0.269
Age	0.307	0.525	0.094	0.185	0.998
Diet × Age	0.378	0.211	0.948	0.024	0.134

^a,b^ Treatment means lacking a common superscript letter differ (*p* < 0.05). ^1^ Abbreviations: CONT, control diet; NOR, novel object recognition; SEM, standard error of the mean; TEST, test diet. ^2^
*p*-value derived from repeated-measures analysis of variance for the main effects and the interaction. ^3^
*p*-values are derived from log-transformed data due to a violation of the homogeneity of variance assumption.

## Data Availability

The data that support the findings of this study are available from the corresponding author upon reasonable request.

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
