# Peer review of "Influence of Dietary Polar Lipid Supplementation on Memory and Longitudinal Brain Development"

_nutrients, 2021, doi:10.3390/nu13082486_

Round 1

Reviewer 1 Report

This is a very interesting animal study.

The Researchers are experts on the new finding of milk polar lipids and their essential biological functions in brain development and cognition.

The Authors of this work rated, by magnetic resonance imaging, the longitudinal effects on brain macrostructural and microstructural development and recognition memory of early-life polar lipid supplementation using the translational pig model.

From the methodological point of view the study seems adequately appropriate and suitable to scientific rigour; the study design process is valid, both inclusion and exclusion criteria are compliant, as well as data collection; materials and measurement methods are highly specific and sensitive; the results are clearly reported; the statistical analysis seems also valid and the scientific contents are supported by valid References and the narrative is comprehensive and adequately clear/understandable to common Reader; figures and tables are sufficiently clear.

Subtle differences were observed, between intervention and control group, in hippocampal absolute brain volumes (however the hippocampus exhibited a higher rate of development in absolute brain in tested pigs than in control pigs) and fractional anisotropy, and no differences in myelin water fraction developmental patterns were noted. Behavioral outcomes did not differ in recognition memory, and only minimal differences were observed in exploratory behaviors. The Researchers conclude suggesting that early-life dietary supplementation of polar lipids has limited effect on brain developmental patterns, object recognition memory, and exploratory behaviors.

I think this research will be a useful tool for other large scale and human studies and it will stimulate other Researchers.

I have no suggestions to give to the Authors nor specific comments to make; I would just like to advise the Authors to replace the term "subject" with the term "animals" or "pigs", as this is an animal and not human study. 

Author Response

Reviewer 1

I have no suggestions to give to the Authors nor specific comments to make; I would just like to advise the Authors to replace the term "subject" with the term "animals" or "pigs", as this is an animal and not human study. 

Response: We have replaced the term "subject" with the term "animals" or "pigs" throughout the manuscript.

Reviewer 2 Report

Comments to the Author 
The paper “Influence of Dietary Polar Lipid Supplementation on Memory and Longitudinal Brain Development” describes the possible effects of a mixture of polar lipids supplements in the formula to piglets on the brain development and recognition memory in a longitudinal way. The main criticisms of the paper are the design and outcome of this work.

  1. The detailed information of the polar lipids applied in this work is missing.
  2. The evidence why the formulation of these lipids were applied should also be explained in the methods section. Since there are already studies proved the effects of the individual lipid molecules, why this complex mixture was used? The individual lipid molecule might be applied as additional controls as well.
  3. What is the rational for 48h of colostrum before exposure to distinct diet? will this lead to the non-significant results?
  4. The methods section is not clear enough, please add corresponding subtitles to each measure.
  5. The non-significant results keeps disappointing the readers, in addition to the non-invasive measures, have the authors ever thought to include other measures through anatomy or effects on the gut microbiota composition and the metabolism?
  6. Please check the footnotes of Table 3 and Table 4, and format Table 5.

Author Response

Reviewer 2

The detailed information on the polar lipids applied in this work is missing.

Response: We have specified the polar lipids applied in the Introduction lines 95-97 and methods lines 152-154.

The evidence why the formulation of these lipids was applied should also be explained in the methods section. Since there are already studies proved the effects of the individual lipid molecules, why this complex mixture was used? The individual lipid molecule might be applied as additional controls as well.

Response: There were no specific levels chosen for the polar lipid concentration; we sought to include practical levels of polar lipids as provided by a commercially available, enriched source of whey protein isolate, thereby mimicking a 'whole food' approach. The complex mixture was used as many studies evaluate dietary polar lipid health benefits using individual polar lipids or a combination with MFGM. Our objective was to We wanted to observe the influence of a combination of polar lipids that go beyond just MFGM and this sentiment was clarified in the Introduction lines 97-100.

What is the rationale for 48h of colostrum before exposure to a distinct diet? will this lead to non-significant results?

Response: We have included a statement in Section 2.1 lines 121-123 explaining that most formula-fed human infants consume colostrum and would then receive formula thereafter. Thus, we wanted our translational pig model to closely mimic the practical context of human infants and therefore pigs were provided colostrum before receiving the milk replacer diet. Moreover, as acknowledged in our publications, the passive immunity afforded by colostrum intake is critical to permit pigs neonatal pigs to thrive in a setting absent the natural care provided by their dam.

The methods section is not clear enough, please add corresponding subtitles to each measure.

Response: Subtitles have been added for clarification.

The non-significant results keep disappointing the readers, in addition to the non-invasive measures, have the authors ever thought to include other measures through anatomy or effects on the gut microbiota composition and the metabolism?

Response: Yes, we are interested in exploring the potential influences of early-life polar lipid supplementation on serum vitamin and mineral, phospholipid and fatty acid, and amino acid levels, as well as the gut microbiota, and will hopefully be able to analyze these measures sometime in the near future. 

Please check the footnotes of Table 3 and Table 4, and format Table 5.

Response: We have made the necessary changes to Tables 3 and 4 and have formatted Table 5.

Reviewer 3 Report

This manuscript describes the effects of supplementation of infant formula with polar lipids on memory, macrostructural, and microstructural brain development in postnatal pigs. Pigs supplemented with polar lipids demonstrated a small increase in hippocampal volume, with very little difference in memory and exploratory behavior compared to control. This surprising lack of effect may have been due to the lipid formulation or a lack in cognitive stimulation of their living conditions. Also appreciate the effort the authors took to accurately and fully describe their methods, which is rarely done these days.

Minor:

  1. In the Introduction (Lines 46-47), aren’t ‘cell signaling’ and ‘cellular signal transduction’ more or less the same process? If authors think these are differentiated, suggest adding more detail to make that more obvious.
  1. Line 61, ‘Bayely’ should be ‘Bayley’. When this scale is first used (Line 61), the full name should be stated, rather than stating later in Line 65.
  1. In the Introduction, Lines 50-72, authors talk about the benefits of specific lipids, then move more generally to MFGM. I would suggest doing the reverse…talk about the benefits of MFGM, then move to benefits of specific lipids within the MFGM.
  1. Last paragraph of Introduction (Lines 73-86), numerous typos and grammatical errors.
  1. Since the diets were essentially isocaloric, which macronutrient was decreased in order to accommodate the extra calories from the polar lipids, and by how much? Would have been interesting to compare DEXA (or similar) scans to see if body composition differed between the groups, not just body weight.
  1. The authors touched on this limitation a bit in the Conclusion, but I would think having just one enrichment toy in a very small enclosure would not be enough to see drastic changes in neural outcomes. I realize this is the general reality of a farm pig, but wouldn’t the better test be to have the outside exploring, living a more ‘natural’ life? I would think most animals restricted to an artificial enclosure would suffer cognitively.

Author Response

Reviewer 3

In the Introduction (Lines 46-47), aren’t ‘cell signaling’ and ‘cellular signal transduction’ more or less the same process? If authors think these are differentiated, suggest adding more detail to make that more obvious.

Response: Yes, it is a very similar process therefore we removed ‘cellular signal transduction.'

Line 61, ‘Bayely’ should be ‘Bayley’. When this scale is first used (Line 61), the full name should be stated, rather than stating later in Line 65.

Response: Thank you for clarifying, we have made the necessary changes.

In the Introduction, Lines 50-72, the authors talk about the benefits of specific lipids, then move more generally to MFGM. I would suggest doing the reverse…talk about the benefits of MFGM, then move to the benefits of specific lipids within the MFGM.

Response: We appreciate the recommendation and revised the Introduction accordingly.

Last paragraph of Introduction (Lines 73-86), numerous typos and grammatical errors.

Response: We have made the necessary revisions.

Since the diets were essentially isocaloric, which macronutrient was decreased in order to accommodate the extra calories from the polar lipids, and by how much? Would have been interesting to compare DEXA (or similar) scans to see if body composition differed between the groups, not just bodyweight.

Response: The test diet was created by replacing the control whey protein isolate source with another source of whey protein isolate that was specifically enriched in polar lipids and then equalizing amino acid concentrations between the diets. This produced isocaloric diets as we adjusted the inclusion of lactose and dry-fat sources. We agree that another tangential outcome of the study could have been body composition, so we appreciate this comment and will keep this idea in mind for future studies.

The authors touched on this limitation a bit in the Conclusion, but I would think having just one enrichment toy in a very small enclosure would not be enough to see drastic changes in neural outcomes. I realize this is the general reality of a farm pig, but wouldn’t the better test be to have the outside exploring, living a more ‘natural’ life? I would think most animals restricted to an artificial enclosure would suffer cognitively.

Response: The toy in every home cage was to provide a source of enrichment for each animal. Behavior was assessed utilizing the novel object recognition (NOR) paradigm which consisted of placing pigs in a specific arena for behavioral assessment and utilized toys that were different from the enrichment toy in each home cage. NOR was used as it is a behavioral paradigm that has been validated for the pig. Additionally, it is suitable for longitudinal studies as it requires no operant training and relies on the animal’s unconditioned responses to novelty. Therefore, it can be and was used repeatedly on the same animal.